# Baseline Characteristics of Patients Enrolled in Clinical Trials of Biologics for Severe Asthma as Potential Predictors of Outcomes

**DOI:** 10.3390/jcm12041546

**Published:** 2023-02-15

**Authors:** Francesco Menzella

**Affiliations:** Pulmonology Unit, S. Valentino Hospital—AULSS2 Marca Trevigiana, Via Sant’ Ambrogio di Fiera, n. 37, 31100 Treviso, Italy; francesco.menzella@aulss2.veneto.it

**Keywords:** severe asthma, biologics, randomised clinical trials, biomarkers, efficacy

## Abstract

(1) Background: Over the past 20 years, monoclonal antibodies have been developed for the treatment of severe asthma, with numerous randomised controlled trials (RCTs) conducted to define their safety and efficacy. The growing availability of biologics, which until now have only been available for T2-high asthma, has been further enriched by the arrival of tezepelumab. (2) Methods: This review aims to evaluate the baseline characteristics of patients enrolled in RCTs of biologics for severe asthma to understand how they could potentially predict outcomes and how they can help differentiate between available options. (3) Results: The studies reviewed demonstrated that all biologic agents are effective in improving asthma control, especially with regard to reducing exacerbation rates and OCS use. As we have seen, in this regard, there are few data on omalizumab and none yet on tezepelumab. In analysing exacerbations and average doses of OCSs, pivotal studies on benralizumab have enrolled more seriously ill patients. Secondary outcomes, such as improvement in lung function and quality of life, showed better results—especially for dupilumab and tezepelumab. (4) Conclusion: Biologics are all effective, albeit with important differences. What fundamentally guides the choice is the patient’s clinical history, the endotype represented by biomarkers (especially blood eosinophils), and comorbidities (especially nasal polyposis).

## 1. Introduction

Asthma is a disease characterised by a great heterogeneity of immunological pathways and clinical manifestations, and it requires an approach based on personalised medicine and the study of phenotypes and endotypes [1,2]. Over the past 20 years, biological drugs have been developed for the treatment of severe refractory asthma, with numerous randomised controlled trials (RCTs) conducted to define their safety and efficacy [3]. These drugs have targeted mechanisms of action and are addressed to specific patients, albeit there is a certain degree of overlap regarding their inflammatory characteristics—especially those with type 2 (T2)-high asthma. Patients enrolled in the different RTCs conducted during these years often had different characteristics, which influenced the relative results. Knowing these differences can be useful for appropriately interpreting the magnitude of the results and positioning the different therapeutic options in the best possible way. The growing availability of biologics—thus far only available for T2-high asthma—has been further enriched by the arrival of tezepelumab. This anti-thymic-stromal-lymphopoietin (TSLP) biologic is also indicated for both T2-high and T2-low asthma. It is important to understand in detail what kinds of patients were enrolled in the RCTs and their outcomes, even in comparison with already-available biologics [4]. In this regard, many indirect comparisons have been conducted recently, including tezepelumab, as there have been no head-to-head studies [5,6,7]. These analyses mainly considered statistically significant data, overshadowing the clinical aspects and characteristics of the patients enrolled in the trials.

This review aims to evaluate the baseline characteristics of the patients enrolled in RCTs of biologics for severe asthma, to better define the current landscape and understand how they could potentially predict outcomes, how they can help differentiate between available options, and to give useful insights to clinicians for a more appropriate choice.

## 2. Asthma Phenotyping for Correct Patient Selection

The choice of biologics is crucial in the management of patients with severe uncontrolled asthma. In the literature, the presence of a mixed phenotype varies by 37–47%, depending on the case [8,9]. This can lead to an overlap in eligibility for biological treatment, which is why it is important to precisely identify the phenotype, given that many monoclonal antibodies can be used in various types of patients (Table 1). Therefore, it should be clarified whether patients have early-onset allergic asthma or late-onset eosinophilic asthma [10]. The presence or absence of manifestations or comorbidities correlated with atopy or eosinophilia is also important, such as allergic rhinitis in the first case and chronic rhinosinusitis with nasal polyps (CRSwNP) in the second. The first option considers anti-immunoglobulin E (IgE) as the first choice, even in the presence of contemporary eosinophilia. In the second hypothesis, anti-interleukin-5 (IL-5) or IL-5 receptor alpha (Rα), dupilumab, or tezepelumab should be used, but omalizumab is always chosen due to its long experience in terms of safety and efficacy. Omalizumab and tezepelumab can also be used for patients with late-onset allergic asthma with sensitisation to perennial allergens and without eosinophilia. Anti-IL-5/IL-5Rα can be used in early-onset allergic asthma with eosinophilia. In patients with late-onset allergic asthma and eosinophilia, it is important to evaluate the presence of eosinophilic manifestations, such as CRSwNP, aspirin exacerbation respiratory disease, and atopic dermatitis. If these are not present, omalizumab may be chosen; otherwise, IL-5/IL-5Rα or dupilumab is recommended [11]. In patients with non-allergic eosinophilic asthma, omalizumab is not indicated, and anti-IL-5/IL-5Rα, dupilumab, or tezepelumab should be considered. As direct comparison studies are lacking, we must keep in mind the predictive factors of responses evaluated in clinical studies, meta-analyses, post hoc studies, and real-world studies. These factors may be represented by blood eosinophil count, fractional exhaled nitric oxide (FeNO), baseline oral corticosteroid (OCS) use and dosage, symptoms, exacerbations, respiratory function, and the presence and type of comorbidities [11,12]. To help clinicians, some authors have developed a score based on forced expiratory volume in the first second (FEV_1_), exacerbations, OCSs, and symptoms, which can quantify the response to biologics [12]. These parameters, together with others such as baseline demographic characteristics and biomarker levels, are heterogeneous when carefully evaluating the different phase 3 RCTs of biologics. This has great importance in the interpretation of related outcomes.

## 3. Results of Major Clinical Trials

Five biologics are currently available for the treatment of severe allergic (omalizumab), eosinophilic (mepolizumab, reslizumab, and benralizumab), and mixed asthma with atopic–eosinophilic phenotype (dupilumab), along with tezepelumab [13]. In the United States, the latter has been available since 2022. Tezepelumab has also been approved in Europe by the European Medicines Agency, and its marketing is expected in 2023. We will not discuss reslizumab in this article because it is only marketed in a few countries. These monoclonal antibodies (mAbs) have common aspects, but also important differences regarding their inclusion criteria—baseline demographic characteristics, and biomarker levels, which are heterogeneous when carefully evaluating the different phase 3 RCTs of biologics (Table 2). These aspects have great importance in the interpretation of related outcomes. Therefore, it is necessary to have a thorough understanding of these differences in biological therapies to avoid simple comparisons and considerations between the different biologics. The different biologics underwent clinical development with important pivotal phase 3 studies characterised by criteria and endpoints with sometimes significant differences, and for each of these open-label extension studies, OCS-sparing studies (except for omalizumab) and further clinical development studies for other indications have been conducted. International regulatory bodies have approved biologics for severe asthma, but regional differences and varying prescription criteria can limit the accessibility of these medicines and pose challenges for precision medicine. National prescription criteria for biologics were also evaluated by a collaborating panel of experts in the International Severe Asthma Registry [14]. The results were based on country-specific prescription criteria and the development of biological accessibility scores. These scores showed substantial differences between countries in terms of ease of access to biologics.

### 3.1. Omalizumab

Omalizumab is a recombinant humanised mAb IgG1 anti-IgE that binds to circulating IgE, preventing its binding to the receptors of mast cells and basophils, and blocking the release of histamine and other mediators [15,16] (Figure 1). It is used as a treatment for patients over 6 years of age with moderate severe allergic asthma (SAA), with serum IgE levels of 30–1300 IU/mL in the United States, or with SAA with serum IgE levels of 30–1500 IU/mL in Europe (Table 3). The dose depends on body weight and serum IgE level, ranging from 150 mg every four weeks (Q4W) to 375 mg every two weeks (q2w) subcutaneously (SC) [17]. Omalizumab is also indicated for the treatment of chronic spontaneous urticaria, with a single dose of 300 mg Q4W [17], and for recalcitrant CRSwNP, with a dosage schedule similar to that for asthma [18]. Over nearly 20 years, numerous studies have examined the efficacy of omalizumab in patients with severe allergic asthma. In these studies, the enrolled patient populations differed somewhat in the definition of severity considered and concomitant medications used, with a trend in the most recent studies towards more severe forms of asthma and more aggressive concomitant therapies. Unlike the mAbs that arrived later, the RCTs on omalizumab did not evaluate the OCS-sparing effect at their endpoints. Data were later obtained from numerous real-life studies conducted over time [19]. In two double-blind, phase 3 RCTs conducted by Solèr et al. [20] and Busse et al. [21], patients aged >12 years with symptomatic allergic asthma were enrolled, despite doses of beclomethasone dipropionate ranging from 500 μg/day to 1200 μg/day. After 28 weeks of therapy, the reduction in inhaled corticosteroid (ICS) use was significantly greater with omalizumab than with the placebo. In these two studies, patients on active treatment had up to 58% fewer asthma exacerbations than those on placebo, and a greater likelihood of ICS dose reduction or discontinuation was observed among patients on active omalizumab treatment compared with placebo. A third pivotal trial—the Investigation of Omalizumab in Severe Asthma Treatment (INNOVATE)—evaluated the efficacy of omalizumab in patients with uncontrolled asthma, despite GINA phase 4 therapy [22]. The study included 419 subjects over 12 years of age with a proven allergy to at least one perennial allergen, a deterioration in respiratory function (predicted FEV_1_ of 40–80%), and a recent clinically significant history of exacerbations despite high doses of ICSs, long-acting beta-agonists (LABAs), and other control agents. The patients had received either omalizumab or a placebo for 28 weeks. During this treatment period, the rate of clinically significant exacerbations was 26% less in the active treatment arm compared with the placebo group (*p* = 0.0002). Severe asthma exacerbations and emergency department visits were significantly reduced, and asthma symptoms and morning peak expiratory flow were significantly improved. EXCELS was a post-marketing observational cohort study undertaken by the Food and Drug Administration (FDA) to evaluate the long-term safety of omalizumab [23]. This study aimed to explore the potential association between omalizumab and cardiovascular (CV) or cerebrovascular (CBV) events. The patient cohort included 5007 active treatment subjects and 2829 controls followed for ≤5 years. The two cohorts had similar baseline demographics, but severe asthma was more frequent in the omalizumab-treated group than in the control group (50% vs 23%). Actively treated patients had a higher rate of serious adverse CV/CBV events (13.4 per 1000 person-years (PY)) than non-omalizumab-treated patients (8.1 per 1000 PY). The differences in asthma severity between the cohorts likely contributed to this imbalance, but some increase in risk could not be ruled out. In light of these considerations, the FDA still did not recommend any changes to the prescription information, recommending only increased awareness. In a 32-week registry-based RCT of 400 patients treated with omalizumab add-on, persistence response was defined using the physician’s global evaluation of treatment efficacy (GETE) [24]. Patients on the optimised standard therapy showed less persistence of response compared with those in the omalizumab group. Good and excellent GETE scores in the omalizumab group were correlated with improvement in exacerbation rates (*p* < 0.001), severe exacerbations (*p* = 0.023), hospitalisations (*p* = 0.003), and overall scores on the asthma control questionnaire (ACQ) (*p* < 0.001). Finally, 62.7% of patients in the active treatment group had reduced or stopped using OCSs, compared with 30.4% in the control arm. The role of combined biomarkers as possible predictors of treatment response was explored in the EXTRA study, which included 850 patients with severe allergic asthma [25]. Outcomes were evaluated in relation to FeNO, blood eosinophil count (BEC), and serum periostin at baseline, and the subgroups with high and low levels of these biomarkers were analysed. During 48 weeks of treatment, the reduction in exacerbations was significant in the subgroup with high biomarkers. A 24-week RCT explored the efficacy of omalizumab on the exacerbation rate [26]. In this study, patients with BEC of 300 cells/µL or greater experienced a 59% reduction in the rate of exacerbations with omalizumab compared with the placebo (0.25 vs. 0.59).

### 3.2. Mepolizumab

Mepolizumab is an anti-IL-5 IgG1k mAb approved for severe eosinophilic asthma (SEA) (Figure 1). Numerous RCTs have confirmed its effectiveness in terms of reducing the rate of asthma exacerbations, as well as providing an OCS-sparing effect, improvement in lung function, and increase in health-related quality of life (HRQoL). The approved dose of mepolizumab (100 mg every 30 days SC) was identified based on the results of the DREAM and MENSA efficacy and safety trials [27,28], and as a steroid-sparing agent (SIRIUS) [29] (Table 3). These RCTs showed that mepolizumab was more effective in patients with BEC greater than 150–300 cells/μL. In a post hoc analysis of the DREAM study, a single BEC measurement of 150/μL or greater predicted the mean of subsequent measurements to be 150/μL or greater in 85% of this population. Using an average of several measurements increased the sensitivity only marginally. Sputum eosinophils did not predict the response to mepolizumab treatment [30]. A secondary analysis of the DREAM and MENSA studies demonstrated that the reduction in exacerbation rate with mepolizumab versus placebo increased progressively from 52% in patients with a baseline BEC of at least 150 cells/μL to 70% in patients with a baseline BEC of at least 500 cells/μL [31]. Additionally, in subjects with a BEC of less than 150 cells/μL, the efficacy of mepolizumab was reduced. This study showed a close relationship between baseline BEC and the clinical efficacy of mepolizumab, with efficacy increasing progressively with increasing BEC. Another RCT, MUSCA, confirmed the efficacy of mepolizumab in significantly improving HRQoL in patients with SEA, with a safety profile comparable with a placebo [32]. OSMO was a 32-week, open-label, single-arm, multicentre trial in which patients already on omalizumab treatment (average of 29 months) but with inadequate response, eosinophilia ≥ 150 cells/µL (or ≥300 cells/µL in the previous year), and an ACQ-5 score ≥ 1.5 had discontinued omalizumab and started mepolizumab with no washout period [33]. These patients then had clinically significant improvements in asthma control, health status, and exacerbation rate, with no safety concerns. The extension studies (COSMOS, COLUMBA, and COSMEX) confirmed the long-term safety and tolerability of mepolizumab for up to 4.8 years [34,35,36].

### 3.3. Benralizumab

Benralizumab is an IgG1 mAb targeting the α subunit of the IL-5 receptor. It activates an antibody-mediated cellular cytotoxicity mechanism, leading to a profound depletion of eosinophils and their precursors in the blood and tissues [37] (Figure 1). In the Windward development programme, two phase 3 studies (CALIMA and SIROCCO) explored the efficacy of benralizumab in patients with high eosinophil levels (≥300 cells/μL) and low eosinophil levels (<300 cells/μL) [38,39] (Table 3). In the CALIMA study, benralizumab reduced asthma exacerbations by 36% and 28% compared with a placebo in the high-BEC population in groups dosed every 4 and 8 weeks, respectively. SIROCCO showed superior performance, as benralizumab reduced the exacerbation rate by 45% and 51% compared with the placebo in the high-BEC population dosing every 4 (Q4W) and 8 weeks Q8W), respectively. In the low-BEC population, the reduction in the exacerbation rate was only 17% compared with placebo Q8W. A pooled analysis of the results from the SIROCCO and CALIMA studies found that the greatest improvements in the annual exacerbation rate (AER) compared with placebo were seen in patients with a combination of high BEC (≥300 or ≥450 cells/μL) and a history of more frequent exacerbations (three or more) [40]. In the phase 3 ZONDA study, the primary endpoint was the percentage change in OCS dose from baseline to treatment week 28 [41]. Benralizumab showed a 50% reduction in prednisone dose compared with the placebo. In the secondary outcomes, Q8W administration resulted in a 55% lower annual exacerbation rate compared with the rate with placebo (*p* = 0.003), and benralizumab Q8W resulted in a 70% lower annual exacerbation rate than placebo (*p* < 0.001). At the end of the study, no significant effect on FEV_1_ was found. A post hoc analysis evaluated the rate of patients in clinical remission among pooled patients in the SIROCCO/CALIMA or ZONDA trials [42]. In the first two trials, 14.5% (85/586) of patients treated with benralizumab and 7.7% (48/620) of those on placebo achieved clinical remission at 12 months. Examining the patients enrolled in ZONDA, 22.5% (9/40) of the active benralizumab group achieved clinical remission, compared with 7.5% of the placebo group. An interesting open-label, single-arm, multicentre study had, as its primary endpoints, a proportion of patients able to eliminate sustained daily OCS use for at least 4 weeks, and a proportion achieving weaning from OCSs or a daily dose of prednisone or prednisolone of 5 mg or less for at least 4 weeks if the reason for complete discontinuation was adrenal insufficiency (AI) [43]. Moreover, 376 (62.88%) of 598 patients eliminated OCSs, and 490 (81.94%) of 598 patients eliminated their use or achieved a dose of 5 mg or less if the reason for incomplete elimination was AI. When examining subgroups, dose reductions were achieved regardless of the BAC and baseline OCS dose or the duration of treatment with them. AI was detected in 60% of patients in the first evaluation, and in 38% three months later. This is the first and only study to date that has been able to provide useful practical information on the management of OCS tapering in patients with severe asthma treated with biologics. The long-term open-label MELTEMI extension study confirmed the high safety and efficacy of benralizumab with up to five consecutive years of treatment [44].

### 3.4. Dupilumab

Dupilumab is a fully human IgG4 mAb directed to the alpha subunit of the shared receptors of IL-4 and IL-13 (IL-4/13Rα), simultaneously blocking the IL-4/IL-13 signalling pathway [45,46] (Figure 1). The safety and efficacy of the subcutaneous administration of dupilumab for the treatment of moderate-to-severe asthma were primarily demonstrated in three phase 3 clinical studies. The first study enrolled patients aged 12 years and older with BEC ≥ 150 cells/μL and FeNO ≥ 25 parts per billion (ppb). Dupilumab, administered to these patients every two weeks (q2W), was effective in reducing the exacerbation rate, symptoms, and lung function compared with placebo, as well as in reducing FeNO. The second study was a 52-week phase 3 RCT (LIBERTY ASTHMA QUEST) that confirmed the efficacy of dupilumab in terms of a significant reduction in exacerbations and an improvement of FEV_1_ and asthma control, especially in patients with higher T2 biomarkers (BEC ≥ 300 cells/µL and FeNO ≥ 25 ppb) [47] (Table 3). The third study was the LIBERTY ASTHMA VENTURE, which involved a cohort of patients with severe steroid-dependent asthma. Dupilumab at a dose of 300 mg Q2W significantly reduced the rate of severe exacerbations, with a concomitant increase in FEV_1_. Specifically, the mean daily dose of prednisone was reduced by 70%, compared with 42% in the placebo group [48]. In this trial, transient eosinophilia was reported in approximately 14% of patients on active therapy, with normalisation occurring by the end of the treatment period and no patients experiencing clinically relevant adverse events [48]. A recent pooled analysis of the VENTURE study showed that dupilumab significantly reduced the rate of exacerbations and improved FEV_1_, asthma control, and HRQoL, regardless of the presence or absence of atopy [49]. Subsequently, an open-label extension study called TRAVERSE was conducted to evaluate the safety and efficacy of dupilumab over 148 weeks of treatment [50]. As in the parent studies, an increase in BEC was observed after the initiation of dupilumab. The blood eosinophils returned to baseline by week 96, confirming that the increase was transient only, with no evidence of an increased frequency of clinically significant adverse effects compared with previous studies. A recent post hoc analysis of TRAVERSE confirmed dupilumab’s ability to sustain OCS dose reduction while maintaining a low exacerbation rate, improving lung function [51]. In another post hoc analysis of the TRAVERSE study, the long-term efficacy of dupilumab was observed in patients with asthma, both with and without CRSwNP, including the OCS-sparing effect [52]. In a post hoc analysis of 11 clinical trials of dupilumab in patients with moderate-to-severe corticosteroid-dependent asthma, transient increases in the mean blood eosinophil counts were observed in patients with dupilumab-treated asthma (mean range from studies to baseline: 349–370 cells/µL; week 4: 515–578 cells/µL) [53]. A decline was then observed from week 24 to baseline, or even below. No elevations were observed in patients with eosinophilic esophagitis or atopic dermatitis. In all studies, eosinophilia rates ranged from 0% to 13.6%. Clinical symptoms associated with this increase were rare (7 of 4666 dupilumab-treated patients, including 6 cases of eosinophilic granulomatosis with polyangiitis) and occurred only in patients with asthma or CRSwNP. Eosinophilia was not associated with decreased efficacy of dupilumab.

### 3.5. Tezepelumab

A new first-in-class biologic drug has been granted an FDA breakthrough designation for non-eosinophilic asthma: tezepelumab [54]. Tezepelumab is a fully human IgG2 mAb directed to the thymic stromal lymphopoietin (Figure 1)—a cytokine of the alarmin family that, together with IL-25 and IL-33, is derived from epithelial cells and plays a very important role in the pathogenesis of asthma [55]. 

The phase 3 programme called PATHFINDER brought interesting results, but also some queries. The NAVIGATOR trial randomised 1061 patients [56] (Table 3). In this study, the annualised rate of exacerbations was 0.93 in the tezepelumab group and 2.10 in the placebo arm. Regarding respiratory function, the FEV_1_ values were 0.23 L and 0.09 L in the active group and the placebo group, respectively (*p* < 0.001). Among the patients without type 2 inflammation (baseline BEC < 150 cells/μL and baseline FeNO < 25 ppb) enrolled in the NAVIGATOR study, the therapeutic effect of tezepelumab was positive, albeit with values generally lower than those in patients with type 2 biomarkers. However, the effect was considered clinically significant. Unfortunately, in the phase 3 RCT SOURCE, with 150 patients randomised into 210 mg tezepelumab Q4W or placebo for 48 weeks of treatment, the primary endpoint of the OCS-sparing effect compared with the control group was not met [57]. A statistically significant reduction was observed only in the subgroup of patients with BEC ≥ 150 cells/μL, and an even more marked reduction in those with ≥ 300 cells/μL. Beyond this, however, the endpoint of the reduction in the use of OCSs regardless of the status of the biomarkers was not achieved—a problem that will require further investigation to clarify whether it was due to an incorrect study design or a limitation of the biologic. In this regard, a new 28-week RCT, with the primary endpoint being the percentage of participants able to discontinue OCS use without losing asthma control, is underway (NCT05274815 WAYFINDER) [58]. We can also report interesting data that emerged from the UPSTREAM study, although this was only a phase 2 trial [59]. In this RCT, 40 adult patients with asthma and an indirect bronchial provocation test with mannitol airway hyperresponsiveness (AHR) were randomised to receive 700 mg of tezepelumab or placebo intravenously Q4W for 12 weeks. The mean change in the mannitol challenge dose resulting in a 15% reduction in the FEV_1_ provocative dosage (PD15) with tezepelumab was 1.9 (95% CI 1.2–2.5) versus 1.0 (95% CI 0.3–1.6) by doubling the doses with placebo (*p* = 0.06). Nine (45%) subjects in the tezepelumab group and three (16%) in the placebo group had a negative PD15 test at week 12 (*p* = 0.04). The airway tissue eosinophil and bronchoalveolar lavage levels decreased by 74% and 75%, respectively, in the tezepelumab arm, compared with a 28% increase and 7% decrease in the placebo arm (*p* = 0.004 and *p* = 0.01), respectively. This study led to an editorial in which it was noted that an even greater effect could have been achieved if only patients with severe mannitol AHR were recruited [60]. In addition, the data suggest that the primary mechanism by which tezepelumab works to improve the clinical and physiologic outcomes of asthma is airway eosinophilia suppression (not to zero), and that there may be an interaction between eosinophils and mast cell activity, which needs further investigation. A secondary analysis was also conducted among patients potentially eligible for omalizumab enrolled in NAVIGATOR [61]. In this subgroup, tezepelumab significantly reduced the annualised asthma exacerbation rate (AAER) at various levels of baseline BEC and FeNO. Tezepelumab also improved FEV_1_ and patient-reported outcomes (PROs) and reduced type 2 biomarkers compared with placebo in all patients, as well as in those with severe allergic asthma. DESTINATION is an ongoing phase 3 RCT involving patients who completed the NAVIGATOR and SOURCE studies [62]. The primary endpoint of this study is the evaluation of long-term safety and efficacy for a period of 104 weeks, including the parent studies. The primary endpoint is adverse events, and the secondary endpoint is AAER throughout the observation period. The interim data show no differences in the overall adverse events, including heart disorders, but an increase in the number of serious heart conditions. As this study is still ongoing and the final results are not yet available, a definitive judgment cannot yet be made. However, it is important to clarify that no causal relationship has been established between tezepelumab and these events, nor has a subpopulation of patients been identified as being at a greater risk.

## 4. Overview of Clinical Trials According to Patient Characteristics

The almost absolute majority of patients enrolled in severe asthma RTCs are primarily classified as having T2-high endotypes (Table 4). At the start of the development programme on tezepelumab, we witnessed the enrolment of patients with T2-low endotypes—an aspect that could respond to an important unmet need. The correct selection process is based on the underlying endotyping mechanism and the corresponding phenotype [63]. The main parameter considered in the selection process of potentially eligible subjects was the rate of exacerbations, followed by OCS dependence. Respiratory function was included in the additive parameters, along with disease control and the use of rescue medications, almost systematically in all studies. An initial analysis indicated that the selection process was driven by the presence of atopy for patients selected for anti-IgE RCTs, the presence of eosinophilic inflammation for mepolizumab and benralizumab, and a T2-high response for dupilumab [63]. Even the presence of some comorbidities can support the decision-making process, as some of them are also possible predictors of response. The most representative were allergic rhinitis in the case of omalizumab and CRSwNP in the cases of mepolizumab, benralizumab, and especially dupilumab. Interestingly, no studies have reported significant safety concerns other than initial concerns about increased blood eosinophilia, which have been dispelled by extension studies and related secondary analyses [50,51,52,53].

### 4.1. Biomarkers

With regard to blood eosinophilia, it is interesting to note how the baseline values differed between pivotal studies, due to the selection of patients based on partially different characteristics and who were at least theoretically more suited to the mechanism of action of the various biological agents (Table 4). The eosinophilia value was 300 cells/µL in the patients enrolled in the omalizumab INNOVATE study [22] and 290 cells/µL in the mepolizumab MENSA study [29]. The baseline BEC was 350 cells/µL in the benralizumab SIROCCO study [39] and 250 cells/µL in both the dupilumab LIBERTY ASTHMA QUEST study [47] and the tezepelumab NAVIGATOR study [56].

It should be noted that the baseline level of BEC affects important outcomes, such as exacerbation for all mAbs examined, even in anti-IgE, which does not target these cells but can also have indirect effects on eosinophils [25,26,64] (Table 5 and Figure 2). Higher BEC values also resulted in a lower exacerbation rate for mepolizumab, as evidenced by post hoc analyses in which the reduction in the exacerbation rate increased progressively from 52% in patients with BEC ≥ 150 cells/μL to 70% in those with a baseline count of ≥ 500 cells/μL [31]. Similar evidence was also found for benralizumab (exacerbation rate of −51% overall and −55% in the subgroup with BEC ≥ 300 cells/µL) [38,39], dupilumab (exacerbations rate of −47% overall and −67% in the subgroup with BEC ≥ 300 cells/µL) [47,48], and tezepelumab, for which eosinophilia was surprisingly the biomarker most predictive of a better response (exacerbation rate of −50% overall and −70% in patients with BEC ≥ 300 cells/µL) [56,57] (Table 5). With regard to this biologic, given that it is the only mAb with an indication for T2-low asthma at the moment, it is interesting to highlight how the presence of eosinophilia enables a better response in terms of the OCS-sparing effect. As previously discussed, the SOURCE study missed its endpoint, except in the subpopulation of patients with BEC ≥ 150 cells/µL (or even more in the group with ≥300 cells/µL) [57]. Having evaluated these data from clinical trials, it is important to point out that there are evident discrepancies in some studies carried out in the context of real-world evidence. In this type of investigation, the presence of high BEC values was not relevant to the outcomes of omalizumab. The STELLAIR and PROSPERO studies had stratified blood eosinophils, demonstrating that the response to omalizumab in asthma control did not change at any level of eosinophilia [65,66]. These studies showed that a large percentage of patients with SAA had a BEC ≥ 300 cells/µL, particularly highlighting that the efficacy of omalizumab was comparable in the ‘high’ and ‘low’ eosinophil subgroups. The REDES multicentre observational real-life study on patients with SEA treated with mepolizumab also confirmed that this anti-IL-5 mAb was effective in reducing the exacerbation rate, improving lung function, and reducing the mean dose of OCSs, regardless of BEC [67]. 

Regarding atopy, all patients had this characteristic in the case of anti-IgE [22], along with 50% of patients enrolled in mepolizumab MENSA [28], 60% in mepolizumab SIROCCO [39], 83% in dupilumab LIBERTY ASTHMA QUEST [47], and 68.6% in tezepelumab NAVIGATOR [56] (Table 4). Baseline IgE was the only predictor of efficacy in the INNOVATE study, but a pooled analysis of seven RCTs showed that the omalizumab response was actually independent of IgE levels [68]. Clearly, the presence of sensitisation to perennial aeroallergens was mandatory. The IgE concentrations and the presence of atopy were not found to influence the efficacy of mepolizumab, benralizumab, and dupilumab, as identified by different studies and post hoc analyses [69,70,71]. Tezepelumab deserves a separate discussion, having been developed for both T2-high and T2-low asthma. As we have seen, the outcomes were achieved in both subpopulations, although the results were better among patients with co-expression of T2 biomarkers. However, blood eosinophils were found to be more important as biomarkers than the presence of atopy in improving clinical efficacy and reducing OCS usage [56,57]. 

FeNO is a useful biomarker for T2-high asthma phenotyping and a predictor of the response to some biologics. Its production is induced by pro-inflammatory cytokines, including tumour necrosis factor-α, IL-1b, IL-4, and IL-13 [72]. The omalizumab EXTRA study confirmed the usefulness of Th2 biomarkers, including high FeNO values (>19.5 parts per billion (ppb)), in predicting response to omalizumab [25]. Other studies have also supported this evidence, highlighting how patients with high FeNO values responded better to omalizumab, which maintains its efficacy over time [73]. The clinical efficacy of anti-IL-5/IL-5Rα mAb biologics is independent of the basal levels of FeNO, consistent with the DREAM study and evidence from other authors [27,74]. Therefore, this biomarker is not useful in predicting the response to mepolizumab or benralizumab. However, treatment with these mAbs—especially benralizumab—is associated with a reduction in FeNO, suggesting that IL-5R-expressing cells, including eosinophils and basophils, are a possible source of IL-13 [74]. Increased FeNO (>25 ppb) was associated with a reduction in the annual exacerbation rate and improved lung function in dupilumab-treated patients [47]. In addition, patients with concomitant FeNO values > 25 ppb and BEC > 150, or even better than 300 cells/mmc, showed the greatest clinical benefits in terms of the OCS-sparing effect [48] (Table 5 and Figure 2). In a post hoc analysis of LIBERTY ASTHMA QUEST, the exacerbation rate and FEV_1_ improved with higher baseline FeNO values and increased with higher baseline FeNO (FeNO less than 25, 25–50 and 50 and greater ppb subgroups). These results were independent of BEC, confirming that FeNO was the main biomarker for the management of patients treated with dupilumab [75]. Concerning tezepelumab, as previously discussed, reductions from the baseline in AAER were observed regardless of the baseline T2-high and -low inflammatory status, assessing BECs, FeNO, total serum IgE, IL-5, IL-13, periostin, and other cytokines [76]. Therefore, the most important biomarker for tezepelumab is eosinophil, as demonstrated by the CASCADE study, in which improvements in the clinical outcomes of asthma were largely driven by the reduction in eosinophilic inflammation, whose reduction was independent of the baseline BEC, but not of the FeNO values [77].

### 4.2. Disease Severity, OCS Use, and Respiratory Function

Examining the baseline patient characteristics of patients enrolled in clinical trials, they differed—at least in part—not only with regard to biomarkers, but also in the level of disease severity. This resulted in different exacerbation rates and prednisone doses, while FEV_1_ and ICS/LABA doses were similar [68]. Focusing on the exacerbations and starting chronologically from omalizumab (INNOVATE) [22], at baseline, the exacerbations were 2.1 in the 12 months before enrolment, 2.8 in the mepolizumab MENSA study [28], 3.8 in the benralizumab SIROCCO study [39], 2.0 in the dupilumab LIBERTY ASTHMA QUEST study [47], and an average of 2 in the NAVIGATOR study before treatment with tezepelumab [56]. Thus, the patients enrolled in SIROCCO were those with the highest number of exacerbations at baseline, confirming a greater severity of asthma or, in any case, a worse response to usual care at the maximal dose.

In analysing steroid use, as pivotal omalizumab clinical trials were conducted many years ago, no specific OCS assessment was performed. For this reason, an average baseline dose of prednisone was not reported in INNOVATE. An OCS-sparing study (EXALT) was subsequently conducted [78]. Eighty-two patients were receiving maintenance prednisone at baseline (mean dose 13.1 mg). The OCS dose reduction was significantly greater in the active treatment group (*p* = 0.002), in which 62.7% reduced or were weaned off OCSs. In the MENSA study, the mean dose of OCSs was 12.6 mg, while the mean dose of patients enrolled in SIROCCO was higher (15.2 mg). No data were available for the LIBERTY ASTHMA QUEST and NAVIGATOR studies because their endpoints were completely delegated to OCS-sparing RCTs. In the dupilumab LIBERTY ASTHMA VENTURE and tezepelumab SOURCE studies, the mean baseline doses were 11 mg and 11.8 mg, respectively [48,57]. When comparing the related steroid-sparing trials of mepolizumab (SIRIUS) and benralizumab (ZONDA), the mean baseline dose in both studies was 10 mg [29,41], although 14% of patients were weaned from OCSs in the SIRIUS study and 52% in ZONDA, which supports a superior OCS-sparing effect of benralizumab compared to mepolizumab [79]. The mean baseline dose of OCS was similar in all studies, and the results obtained were similar for the different biologics (an average reduction of 50% from the baseline), except for tezepelumab, as the SOURCE study did not reach its primary endpoint.

In all investigations, respiratory function was considered as a secondary outcome. Nevertheless, it was still interesting to evaluate the overall behaviour of the biologics analysed. The baseline FEV_1_ reported in the INNOVATE study was predicted to be 61% (absolute value not available). At the end of the treatment period, an improvement of 190 mL was obtained. In the MENSA study, the baseline FEV_1_ value was 1.73 L, and an improvement of 98 mL was reported at the end of the study. The SIROCCO study had a baseline value of 1.65 L, with an improvement of 159 mL. The QUEST trial reported an FEV_1_ value of 1.70 L, with an improvement of 340 mL after treatment. The NAVIGATOR study had a baseline FEV_1_ of 1.8 L, with an improvement of 230 mL.

These results show that the efficacy of mAbs differs in the improvement of respiratory function. Dupilumab was the biologic that obtained the best data, followed by tezepelumab (for the latter, a good result was also reported in terms of the reduction in bronchial hyperreactivity, as previously discussed) (Table 5 and Figure 3). In general, however, all biologics showed a good improvement in FEV_1_, confirming the effectiveness of these agents in this respect.

### 4.3. Comorbidities

Comorbid conditions in severe asthma are common. They complicate management and may affect patient outcomes by contributing to poor disease control. For this reason, identifying comorbidities is important. Comorbidities are numerous, can be pulmonary or extrapulmonary, and are more or less frequently associated with asthma. Among the most common are gastroesophageal reflux disease, bronchiectasis, obesity, allergic rhinitis, depression, diabetes mellitus, and cardiovascular diseases. The most evaluated comorbidity in clinical trials was chronic rhinosinusitis with CRSwNP [80]. All of the biologics discussed here have also been studied for this pathology and, among them, omalizumab, mepolizumab, and dupilumab have obtained indications for treatment, given the good outcomes obtained from the pivotal trials [81]. To date, there are no data from RCTs regarding the possible benefits of omalizumab in asthma and concurrent CRSwNP. Pivotal studies concerned only CRSwNP [18], and many real-life investigations have been conducted. We do not discuss this here because it is beyond the scope of this review. For mepolizumab, a post hoc analysis of the phase 2b/3 DREAM, MENSA, SIRIUS, and MUSCA studies showed that it reduced exacerbations and improved asthma control, HRQoL, and lung function in patients with comorbidities, such as CRSwNP, cardiovascular disease, obesity, gastroesophageal reflux, and diabetes mellitus [32,82]. For this biologic, in addition to the excellent results of pivotal studies on CRSwNP [83], many data have been obtained from real-world studies on asthma and concomitant CRSwNP [84]. Dupilumab is the first mAb to gain indication for CRSwNP, and it is perhaps the one that has shown the best results in the treatment of this disease [85]. A post hoc analysis of the LIBERTY ASTHMA QUEST study was conducted to define the efficacy and safety of dupilumab in patients with uncontrolled, moderate-to-severe asthma with or without self-report of comorbid chronic rhinosinusitis (CRS or non-CRS) [86]. Dupilumab at doses of 200 mg and 300 mg reduced annualised severe exacerbation rates by 63% and 61%, respectively, in patients with CRS, and by 42–40% in patients with asthma without CRS (*p* < 0.001 vs. placebo). In addition, dupilumab improved lung function, asthma control, and HRQoL by suppressing type 2 biomarkers compared with placebo in both subgroups. Benralizumab did not satisfy the outcomes in the pivotal phase 3 OSTRO study in patients with CRSwNP, because it failed to improve the score of the Sinonasal-22 (SNOT-22) outcome test during the first surgery, and the use of OCSs was not statistically significant among the treatment groups [87]. However, a pooled analysis of the phase 3 SIROCCO and CALIMA studies showed that in patients with asthma with a baseline BEC ≥ 300 cells/μL, the use of OCSs, the presence of CRSwNP, a pre-bronchodilator forced vital capacity < 65% of predicted, and ≥3 exacerbations in the previous year were correlated with a greater reduction in exacerbations and increased lung function [88]. The recent phase 3b ANDHI study assessed the effectiveness of benralizumab in terms of the onset of the effect, and of the impact on HRQoL, exacerbation rate, lung function, and symptoms of nasal polyposis (NP) [89]. The subset of patients with NP showed a superior effect of benralizumab in terms of yearly exacerbation reduction (−69% in the NP group and −49% in the overall population) and lung function (FEV_1_ + 320 mL in the NP group and + 160 mL in the overall population). This study confirmed the efficacy of anti-IL-5Rα in patients with NP but, as there were no statistically significant improvements in the SNOT-22 scores and the other outcomes described, the FDA requested additional data. For this purpose, a second phase 3 trial called ORCHID (ClinicalTrials.gov Identifier: NCT04157335) is underway, the main endpoints of which are nasal polyp burden and patient-reported nasal blockage [90]. Preliminary results are expected in the second half of 2024. Tezepelumab is still under development, and its indication for CRSwNP has not yet been registered. However, a post hoc analysis of the phase 2 PATHWAY study found that 210 mg of tezepelumab reduced AAER relative to placebo to a similar extent in both NP+ and NP− patients (75% and 73%, respectively) [91]. A subsequent exploratory analysis also evaluated the effects of tezepelumab in NP+ or NP− patients enrolled in the phase 3 NAVIGATOR study. The analysis showed that tezepelumab achieved an 86% reduction in AAER in NP+ patients and 52% in NP- patients compared to the placebo [92]. To confirm the efficacy and safety of tezepelumab in patients with CRSwNP, the WAYPOINT trial (ClinicalTrials.gov Identifiers: NCT04851964), whose primary endpoints are the change from baseline in total NPS and nasal congestion score, is ongoing. Preliminary results are expected in the summer of 2024 [93]. Currently, there are no head-to-head comparative studies in the literature evaluating the efficacy of different agents. This gap will be filled by the EVEREST study (EValuating trEatment RESponses of dupilumab vs omalizumab in Type 2 patients) (ClinicalTrials.gov Identifiers: NCT04998604) [94]—a phase 4 trial with a primary objective of evaluating the efficacy of dupilumab versus omalizumab in reducing nasal polyp size and improving the sense of smell. Preliminary data will be available in the second half of 2023.

## 5. Conclusions

Over the last 15 years, we have seen a radical change in the management of severe asthma due to the arrival of more options represented by biologics. These biologics have all been focused on the treatment of the T2-high endotype until the advent of tezepelumab, which addresses a large unmet need in patients with T2-low asthma. Fortunately, all randomised controlled trials demonstrated that all mAbs are effective in improving asthma control, especially with regard to reducing the exacerbation rate and the use of OCSs. As we have seen, in this regard, for various reasons, there are few data on omalizumab—and none yet on tezepelumab, as the SOURCE study missed its primary endpoint. Fortunately, a new trial is ongoing that could resolve this important shortcoming [58]. However, it is important to consider the baseline characteristics of the patients enrolled in the various RCTs to understand the level of severity of asthma at the index date. In analysing the exacerbations and the average dose of OCSs, pivotal studies on benralizumab have enrolled more seriously ill patients—an aspect that should be considered in evaluating clinical outcomes (Table 4). Secondary outcomes, such as improvement in lung function and HRQoL, also improved for all agents, albeit with variable results—especially for dupilumab and tezepelumab—in the improvement of FEV_1_. Carefully defining the real efficacy of the different drugs and their correct positioning can also be very useful in the case of overlaps, which can potentially make a patient eligible to receive more than one mAb. In this regard, identifying the predominant mechanism, key biomarkers, and comorbidities is the winning strategy. A history of atopy with sensitisation to perennial aeroallergens and early-onset asthma may predict response to omalizumab. For this purpose, the total IgE level is a biomarker that is useful only for calculating the personalised dose, but not for predicting the response [24]. With regard to blood eosinophils and FeNO, the results were contradictory, and an indication of the measurement of the driving inflammatory process emerged, rather than a possible response prediction [25,65]. The baseline blood eosinophil level can predict response—especially for mepolizumab and benralizumab, and partially for dupilumab, although in the latter case the value of FeNO is very important. Baseline BEC appears to predict the future exacerbation rate response for mepolizumab and benralizumab [31,40,88]. For dupilumab, it is more important to evaluate blood eosinophils and FeNO in combination than as individual biomarkers [47,48]. This information can also be useful in the case of therapeutic shifts, which should in any case be limited as much as possible by applying the available information correctly. Tezepelumab is the first biologic to be cleared for T2-high and T2-low asthma, and it has been shown to be effective for both endotypes. However, even for this drug, the presence of a T2-high inflammatory process allowed superior results to be obtained [61]. This confirms the fact that, in the majority of cases, severe refractory asthma is an inflammatory disease that is predominantly driven by sustained T2 inflammation [95].

Due to the lack of head-to-head studies, a few meta-analyses have been conducted in recent years to compare the efficacy of biologics. Without forgetting the limitations of this type of study, generally only slight differences in efficacy and safety have emerged between the various agents [7,96]. A very recent study based on target trial emulation (emulating a hypothetical randomised study) looked at the possible efficacy of omalizumab, mepolizumab, and dupilumab [97]. In the subpopulation of patients with asthma, BEC ≥ 150 cells/µL, and total IgE 30–700 kU/L, dupilumab showed a greater reduction in exacerbation rate and improvement in FEV_1_ compared to omalizumab and mepolizumab. Not only is it difficult to compare the efficacy of mAbs, the very development of these agents has also always been very complex, as demonstrated by the numerous biologics that have failed during RCTs [98].

A fundamental aspect in considering the process of choosing the correct therapeutic option is the type of comorbidity present, which is closely associated with the type of inflammatory process. Atopy and allergic rhinitis guide the choice and efficacy of omalizumab [99]. Conversely, CRSwNP is crucial in considering a potential response to dupilumab, mepolizumab, and omalizumab, which are also the three biological agents to have obtained authorisation for use in this case. Benralizumab and tezepelumab do not yet have a prescription indication for NP, although many data support the efficacy of these agents in patients with this condition—which, when associated with asthma, makes the benefits even better (similar to dupilumab, for example) [88,92]. It is hoped that ongoing RCTs will also lead to authorisation of these two biologics for the treatment of CRSwNP [90,93]. In conclusion, all biologics are effective from many points of view, and they have some similarities, but just as many differences. Consequently, the approach to correctly choose from the various therapeutic and management options for patients with severe asthma should be multifactorial. What fundamentally guides the choice is the patient’s clinical history, the endotype represented by the different biomarkers (especially from blood eosinophils), and the comorbidities (especially CRSwNP).

## 6. Future Directions

Despite the great advances that have been made in the knowledge of molecular mechanisms, many questions about severe asthma remain open and unanswered, especially regarding the profound understanding of the inflammatory microenvironment and its contributions to clinical expression. Asthma is an increasingly common disease, and severe refractory asthma in particular has a high impact worldwide. For this reason it is necessary to further improve its management [100]. In the near future, the omics approach could be of great help for a more accurate interpretation of endotypes, for the design of new clinical studies, and for the correct positioning of the available options. The task of clinicians is becoming increasingly difficult. The numerous data now available to us can already support the management of these delicate and complex patients, but the availability of new technologies and predictive biomarkers can certainly make the approach even more precise and enable the coveted clinical remission of severe asthma and related comorbidities in more and more patients.

## Figures and Tables

**Figure 1 jcm-12-01546-f001:**
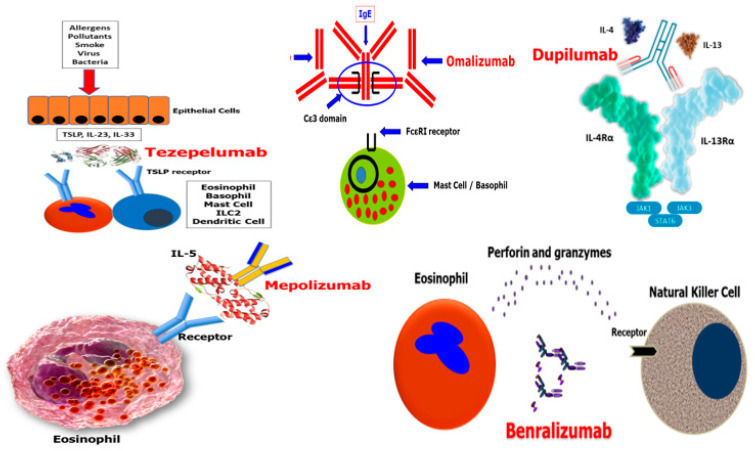
At-a-glance molecular mechanism of action of biologics.

**Figure 2 jcm-12-01546-f002:**
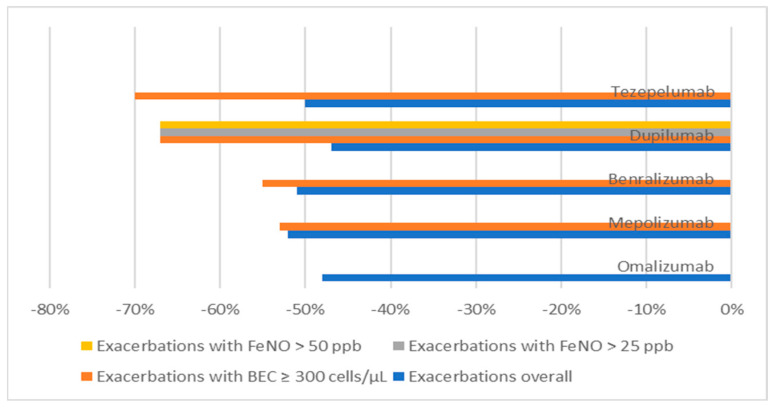
Exacerbations based on biomarkers. FENO: fractional exhaled nitric oxide; BEC: blood eosinophil count.

**Figure 3 jcm-12-01546-f003:**
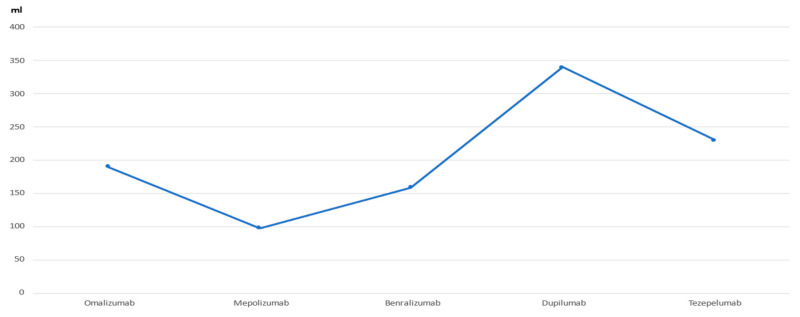
FEV_1_ improvement. FEV_1_: forced expiratory volume in the first second.

**Table 1 jcm-12-01546-t001:** Overview of prescription information and characteristics.

	Omalizumab	Mepolizumab	Benralizumab	Dupilumab	Tezepelumab
Mechanism of action	Anti-IgE	Anti–IL-5 mAb	Anti–IL-5Rα mAb	Anti–IL-4Rα mAb	Anti-TSLP mAb
Asthma US label	Add-on maintenance treatment of adult and adolescent patients (12 years of age and above) with severe persistent allergic asthma who have a positive skin test or in vitro reactivity to a perennial aeroallergen and who have reduced lung function (FEV_1_ < 80%)	Add-on maintenance treatment of adult and pediatric patients aged ≥6 years with severe asthma and with an eosinophilic phenotype	Add-on maintenance treatment of patients with severe asthma aged ≥12 years, and with an eosinophilic phenotype	Add-on maintenance treatment in adult and pediatric patients aged ≥6 years with moderate-to-severe asthma characterized by an eosinophilic phenotype or with OCS dependent asthma	Add-on maintenance treatment of adult and pediatric patients aged ≥12 years with severe asthma
Asthma EU label	Add-on maintenance treatment of adults and pediatric patients 6 years of age and older with moderate to severe persistent asthma who have a positive skin test or in vitro reactivity to a perennial aeroallergen and whose symptoms are inadequately controlled with inhaled corticosteroids	Add-on treatment for severe refractory eosinophilic asthma in adults, adolescents and children aged ≥6 years	Add-on maintenance treatment in adult patients with severe eosinophilic asthma inadequately controlled despite high-dose inhaled corticosteroids plus long-acting β-agonists	Add-on therapy in adults and adolescents aged ≥12 years who have severe asthma with type 2 inflammation characterized by raised blood EOS and/or raised FeNO, who are inadequately controlled with high-dose ICS plus another medicinal product for maintenance treatment	Add-on maintenance treatment of adults and adolescents (12 years of age and older) with severe asthma that is not adequately controlled by a combination of high-dose corticosteroids taken by inhalation plus another asthma medicine
Dosing in asthma	The recommended dosage for treatment of asthma is 75 mg to 150 mg by subcutaneous injection every 2 or 4 weeks based on serum total IgE level (IU/mL) measured before the start of treatment and by body weight (kg) (0.016 mg/kg per IU/mL of IgE)	100 mg SC q4w	30 mg SC q4w (first 3 doses) then q8w	400 or 600 mg SC loading dose, then 200 or 300 mg SC q2w	210 mg SC q4w
Administration	HCP or patient/caregiver (Prefilled syringe)	HCP or patient/caregiver (AI/prefilled syringe)	HCP or patient/caregiver (AI/prefilled syringe)	HCP or patient/caregiver (AI/prefilled syringe)	HCP (Prefilled syringe)
Devices available	▪ Prefilled syringe	▪ Prefilled syringe ▪ Prefilled single-dose AI	▪ Prefilled syringe ▪ Prefilled single-dose AI	▪ Prefilled syringe ▪ Prefilled single-dose AI	▪ Prefilled syringe
Other US- and/or EU-approved indications	▪ Chronic spontaneous urticaria (CSU) ▪ CRSwNP (≥18 years)	▪ CRSwNP (≥18 years) ▪ EGPA ▪ HES	N/A	▪ AD (≥6 months) ▪ CRSwNP (adults) ▪ EoE (≥12 years)	N/A

MAB: monoclonal antibody, FEV_1_: forced expiratory volume in the first second; FENO: fractional exhaled nitric oxide; SC: subcutaneous; HCP: healthcare professional; AI: autoinjector; CSU: chronic spontaneous urticaria; EGPA: eosinophilic granulomatosis with polyangiitis; HES: hypereosinophilic syndrome; CRSwNP: chronic rhinosinusitis with nasal polyps; EOE: eosinophilic esophagitis.

**Table 2 jcm-12-01546-t002:** Clinical development comparison.

	Target	Pivotal Phase 3	Open Label Extension	OCS Sparing	Lung Function	Clinical Development for New Indications	
Omalizumab	Anti IgE	INNOVATE	NA	EXALT	NA	NA	EXPERIENCE: Real-world evidence PERSIST: Real-world evidence
Mepolizumab	Anti-IL-5	MENSA	COSMOS COSMEX COLUMBA	SIRIUS		CHOOSEBETWEENAMAB: Ph4 mepo vs oma REMOMEPO: Airway remodeling	MUSCA: QoL COMET: Discontinuation REALITI-A: Real-world evidence
Benralizumab	Anti-IL-5Rα	SIROCCO CALIMA	BORA MELTEMI	ZONDA	SOLANA	ANANKE: Real-world evidence TATE: Ages 6–11 HAYATE: OCS reduction PROs (BEEPS, POWER, BE-REAL, imPROve) PONENTE: OCS use (open label)	MIRACLE: Med-high ICS/LABA SHAMAL: ICS reduction CHINOOK: Airway remodeling AERFLO: MRI pilot study
Dupilumab	Anti-IL4Rα	QUEST	TRAVERSE TRAVERSE Extension	VENTURE	ATLAS	VESTIGE: Airway remodeling MORPHEO: Sleep disturbances RAPID: Real-world patient registry	EVEREST: Coexisting CRSwNP REVEAL: Real-world patient registry
Tezepelumab	Anti-TSLP	NAVIGATOR	DESTINATION	SOURCE		PATH-HOME: Home use study CASCADE: Ph2 MOA/biopsy study PATHWAY: Ph2	SUNRISE: OCS sparing (double blind, placebo controlled) WAYFINDER: OCS use (open label)

OCS: oral corticosteroid.

**Table 3 jcm-12-01546-t003:** Study design and key inclusion criteria.

Parameter	Omalizumab 75 mg or 150 mg SC q2W or q4W	Mepolizumab 75 mg IV or 100 mg SC q4w	Benralizumab 30 mg SC q4w or q8w	Dupilumab 200 or 300 mg SC q2w	Tezepelumab 210 mg SC q4w
	INNOVATE n = 419	MENSA n = 576	SIROCCO n = 1204	QUEST n = 1902	NAVIGATOR n = 1061
Treatment duration	28 weeks	32 weeks	48 weeks	52 weeks	52 weeks
Study dosing	Oma 75 mg to 150 mg q2W/q4WSC to provide a dose of at least 0.016 mg/kg per IU/mL of IgE	Mepo 75 mg IV or 100 mg SC q4w PBO IV/SC q4w	Benra 30 mg SC q4w Benra 30 mg SC q8w PBO SC q4w	Dupi 200 or 300 mg SC q2w PBO 200 or 300 mg SC q2w	Teze 210 mg SC q4w PBO SC q4W
Patient population	Severe allergic asthma (positive skin prick test to ‡1 perennial aeroallergen and total serum IgE level of ‡30 to 700 IU/mL)	Severe eosinophilic asthma (≥150/uL at BL or ≥300/uL prev 12 mo)	Uncontrolled eosinophilic asthma (no min EOS/FeNO)	Uncontrolled asthma ≥12 mo (no min EOS/FeNO)	Patients aged 12–80 years with severe, uncontrolled asthma
Background medication	High-dose ICS/LABA	High-dose ICS	Medium- to high-dose ICS/LABA for >12 mo	Medium- to high-dose ICS + ≤2 additional controller medications	High-dose ICS ≥1 additional controller medication w/(o) OCS
Key entry criteria					
No. of previous exacerbations	≥2	≥2	≥2	≥1	≥2
Pre-BD FEV_1_, % predicted	▪ 40 to <80% of predicted normal value	▪ <80% (adults) ▪ ≤90% (adolescents)	▪ ≤80% (adults) ▪ ≤90% (adolescents)	▪ ≤80% (adults) ▪ ≤90% (adolescents)	▪ <80% adults ▪ <90% adolescents
Bronchodilator reversibility	▪ 12% from baseline	▪ 12% at visit 1 or 2 or past year, and ≥20% between 2 visits in 12 months	▪ ≥12% and 200 mL in FEV1	▪ ≥12% and 200 mL in FEV1	▪ ≥12% and ≥200 mL in the previous 12 months
Primary end points	▪ AER	▪ AER	▪ AER over 52 weeks ▪ Pre-BD FEV1 change from baseline to week 12	▪ AER ratio	▪ AER over 52 weeks

Q2W: dosed every 2 weeks; Q4W: dosed every 4 weeks; SC: subcutaneous; PBO: placebo; EOS: eosinophils; FENO; fractional exhaled nitric oxide; ICS/LABA: inhaled corticosteroids/long-acting beta-agonists; FEV_1_: forced expiratory volume in the first second; PRE-BD: pre-bronchodilator; AER: annual exacerbation rate.

**Table 4 jcm-12-01546-t004:** Comparison of baseline characteristics.

	Omalizumab (INNOVATE)	Mepolizumab (MENSA)	Benralizumab (SIROCCO)	Dupilumab (QUEST)	Tezepelumab (NAVIGATOR)
Age at diagnosis	44 (12–79)	51 ± 14.5	47 ± 15	48±16	49.9 ± 16.0
Blood eosinophil count (Median)	300 cells/µL	290 cells/µL	360 cells/µL	255 cells/µL	250 cells/µL
Atopy %	100	50	60	83	68.6
Exacerbations	2.1	2.8	3.8	2.0	2.0 (58.7); >2 (41.3)
OCS mg/day Median	N/A	12.6	15.2	N/A	N/A
OCS use %	23	27	17	NA	9.3
FEV_1_	61% of predicted	1.73 L	1.65 L	1.70 L	1.8 L
ACQ	3.0	2.3	2.8	2.7	2.8
CRSwNP%	N/A	14	19	23	17

OCS: oral corticosteroid; FEV_1_: forced expiratory volume in the first second; ACQ: asthma control questionnaire; CRSwNP: chronic rhinosinusitis with nasal polyps.

**Table 5 jcm-12-01546-t005:** Comparison between primary outcomes.

	Omalizumab (INNOVATE)	Mepolizumab (MENSA)	Benralizumab (SIROCCO)	Dupilumab (QUEST)	Tezepelumab (NAVIGATOR)
- Exacerbations (overall)	−48%	−52%	−51%	−47%	−50%
- Exacerbations with BEC ≥ 300 cells/μL	N/A	−53%	−55%	−67%	−70%
- Exacerbations with FeNO > 25 ppb	N/A	N/A	N/A	−67%	−77%
- Exacerbations with FeNO > 50 ppb	N/A	N/A	N/A	−69%	N/A
FEV_1_ improvement (mL)	+190	+98	+159	+340	+230

BEC: blood eosinophil count; FENO: fractional exhaled nitric oxide; PPB: parts per billion; FeNO: forced expiratory nitric oxide; FEV_1_: forced expiratory volume in the first second.

## Data Availability

Narrative review with references retrieved from databases of PubMed, Embase, Scopus, Google Scholar, Google, ScienceDirect, and ISI Web of Knowledge. The references were searched to obtain related literature published in the English language.

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
