# Peer review of "Baseline Characteristics of Patients Enrolled in Clinical Trials of Biologics for Severe Asthma as Potential Predictors of Outcomes"

_jcm, 2023, doi:10.3390/jcm12041546_

Round 1

Reviewer 1 Report

The manuscript is precise and elaborative, while clearly summarizing the role of the monoclonal antibodies in clinical trials for asthma w.r.t to the clinical symptoms shown by the patient. I want to thank the authors for selecting this topic. This paper is very much relevant with the journal's interest. Although the paper contains adequate data but there are few areas in this paper which needs more explanation. Therefore, I recommend a major revision for the manuscript. I explain my concerns in more detail below. I ask that the authors to specifically address each of my comments in their response.

 Major Comments:

·        1. This manuscript lacks novelty. A very similar type of review paper has recently been published in frontiers in immunology on December, 2022 (PMID: 36561741). Both the manuscripts are written in almost same format and also addressing the same issues. In the present manuscript, the authors included patient’s clinical history and try to corelate the usage on monoclonal antibodies with the same. But, In my opinion, this is not enough for the publication of this manuscript. The authors need to at least represent the data in some other format or add some new analysis, so that this publication could be considered for possible publication. They could either add/replace the tables with graphical representation, or add bibliometric analysis or some meta analysis to establish the corelation between patient’s clinical data and the dosage of the drugs.   

Minor Comments:

1. The authors must include latest references like PMID: 36561741, PMID: 35765786 etc., which are missing in the manuscript.

2. The English language at parts are unclear, like in line number 14. The authors are requested to rewrite those sentences in simple language to make it easily understandable for the readers.  

3. The authors need to incorporate discussion on monoclonal antibodies that failed during trails (if any).

Author Response

To the Editor

Thank you for considering our manuscript entitled " Baseline characteristics of patients enrolled in clinical trials on biologics for severe asthma as potential predictors of outcomes".

According to the valuable comments and suggestions of reviewers, we have extensively revised the manuscript and responded point-by-point to the comments, as listed below. We are submitting two versions of the manuscript: one clean revised version and a tracked version with visible amendments.

Yours Sincerely,

The corresponding author, Dr. Francesco Menzella

Point-by-point response #Reviewer 1

Comments and Suggestions for Authors

The manuscript is precise and elaborative, while clearly summarizing the role of the monoclonal antibodies in clinical trials for asthma w.r.t to the clinical symptoms shown by the patient. I want to thank the authors for selecting this topic. This paper is very much relevant with the journal's interest. Although the paper contains adequate data but there are few areas in this paper which needs more explanation. Therefore, I recommend a major revision for the manuscript. I explain my concerns in more detail below. I ask that the authors to specifically address each of my comments in their response.

Major Comments:

  • 1. This manuscript lacks novelty. A very similar type of review paper has recently been published in frontiers in immunology on December, 2022 (PMID: 36561741). Both the manuscripts are written in almost same format and also addressing the same issues. In the present manuscript, the authors included patient’s clinical history and try to corelate the usage on monoclonal antibodies with the same. But, In my opinion, this is not enough for the publication of this manuscript. The authors need to at least represent the data in some other format or add some new analysis, so that this publication could be considered for possible publication. They could either add/replace the tables with graphical representation, or add bibliometric analysis or some meta analysis to establish the corelation between patient’s clinical data and the dosage of the drugs.

Response: We thank the reviewer for the comments. We believe that, although it is not easy for a review to be completely innovative, our manuscript is useful in providing a comprehensive update on the positioning of biologics for severe asthma in the 2023 scenario, providing detailed information and a comparison based on data from pivotal studies that can help clinicians in the selection and management of this complex disease. Unlike ours, the cited study published in Frontiers in immunology (PMID: 36561741) is mainly an overview of the approved biologics and those that have failed the trials and did not discuss in detail the baseline characteristics of the patients (OCS dose, exacerbation rate, etc., respiratory function) and related outcomes (also stratified on the basis of biomarkers) as done in our manuscript.

We have added two new figures with a graphical representation of the main outcomes to make the comparison even more immediate, as suggested, and a figure with the mechanisms of action of the biologicals discussed. We also included some meta-analyses and a very recently published target trial emulation study, as requested.

Minor Comments:

  1. The authors must include latest references like PMID: 36561741, PMID: 35765786 etc., which are missing in the manuscript.

Response: We included and discussed the suggested references  and more in the Conclusion and Future Directions section

  1. The English language at parts are unclear, like in line number 14. The authors are requested to rewrite those sentences in simple language to make it easily understandable for the readers.

Response: The manuscript underwent already in the first version an extensive language editing by a native expert english editor. We have rewritten some sentences in simpler language, as suggested.

  1. The authors need to incorporate discussion on monoclonal antibodies that failed during trails (if any).

Response: The aim of our manuscript is a detailed comparison of outcomes of approved biologics for severe asthma, based on baseline characteristics and biomarkers. We have briefly mentioned in the conclusions that some monoclonal antibodies have failed during the trials, mentioning the suggested review (PMID: 36561741).

Reviewer 2 Report

I read it with great interest, but I have raised several cocerns.

#1. Editing of the tables are needed.

#2. The authors have to draw the graphical figure.

#3. Pease draw the figure of the molecular mechanism of each drug.

#4. This is an excellent hard work.

Author Response

To the Editor

Thank you for considering our manuscript entitled " Baseline characteristics of patients enrolled in clinical trials on biologics for severe asthma as potential predictors of outcomes".

According to the valuable comments and suggestions of reviewers, we have extensively revised the manuscript and responded point-by-point to the comments, as listed below. We are submitting two versions of the manuscript: one clean revised version and a tracked version with visible amendments.

Yours Sincerely,

The corresponding author, Dr. Francesco Menzella

 Point-by-point response #Reviewer 2

I read it with great interest, but I have raised several cocerns.

#1. Editing of the tables are needed.

Response: We edited the tables as requested.

#2. The authors have to draw the graphical figure.

Response: We drew two graphic figures representing the main outcomes (the first on overall exacerbations and stratified on the basis of biomarkers; the second on the improvement of respiratory function).

#3. Pease draw the figure of the molecular mechanism of each drug.

Response: We drew a figure that summarizes the molecular mechanisms of action of the biologicals discussed, as suggested.

#4. This is an excellent hard work.

Response: We thank the reviewer very much for the appreciation.

Round 2

Reviewer 1 Report

The authors tried to incorporate all the modifications suggested. This improves the quality of the manuscript.  

Reviewer 2 Report

I have no further comments.